# Provable Defense against Backdoor Policies in Reinforcement Learning

**Shubham Kumar Bharti**
UW-Madison
Madison, WI, USA
skbharti@cs.wisc.edu

**Xuezhou Zhang**
Princeton University
Princeton, NJ, USA
xz7392@princeton.edu

**Adish Singla**
MPI-SWS
Saarbrücken, Germany
adishs@mpi-sws.org

**Xiaojin Zhu**
UW-Madison
Madison, WI, USA
jerryzhu@cs.wisc.edu

## Abstract

We propose a provable defense mechanism against backdoor policies in reinforcement learning under subspace trigger assumption. A backdoor policy is a security threat where an adversary publishes a seemingly well-behaved policy which in fact allows hidden triggers. During deployment, the adversary can modify observed states in a particular way to trigger unexpected actions and harm the agent. We assume the agent does not have the resources to re-train a good policy. Instead, our defense mechanism sanitizes the backdoor policy by projecting observed states to a 'safe subspace', estimated from a small number of interactions with a clean (non-triggered) environment. Our sanitized policy achieves $\epsilon$ approximate optimality in the presence of triggers, provided the number of clean interactions is $O\left(\frac{D}{(1-\gamma)^4 \epsilon^2}\right)$ where $\gamma$ is the discounting factor and $D$ is the dimension of state space. Empirically, we show that our sanitization defense performs well on two Atari game environments. [1]

## 1 Motivation

The success of reinforcement learning (RL) brings up a new security issue: Often the task is so complex that it takes considerable amount of resources to train a good policy. Such resources are increasingly restricted to large corporations or nation states. Consequently, we imagine in the future many users of RL have to be content with obtaining pretrained policies, without the possibility to (re)train the policy themselves. Downloading a pretrained RL policy from an untrusted party opens up a new attack surface known as backdoor policy attacks.

In a backdoor policy attack, an adversary prepares a pair $(\pi^\dagger, f)$ where $\pi^\dagger$ is a backdoor policy and $f$ is a trigger function. The adversary publishes the backdoor policy $\pi^\dagger$ to be downloaded and used by interested users. $\pi^\dagger$ behaves like an optimal policy under normal deployment, until it is triggered. A naive user, interested in performing well in the underlying RL problem, downloads the backdoor policy $\pi^\dagger$ unaware of the fact that adversary can activate the trigger to make the policy perform poorly in the adversarial environment. For example, $\pi^\dagger$ may be the driving policy for an autonomous car. The car will drive normally under $\pi^\dagger$ until the attacker puts up a special sticker somewhere within the car's camera view. The sticker acts as a trigger; when the trigger is present, the

---

[1]The code available at https://github.com/skbharti/Provable-Defense-in-RL

36th Conference on Neural Information Processing Systems (NeurIPS 2022).

policy $\pi^\dagger$ produces undesirable actions such as crashing the car. Backdoor attacks are well-studied in supervised learning, where instead of a policy $\pi^\dagger$ it is a prediction model that contains built-in backdoors that can be triggered to make wrong predictions [4][11][13]. Backdoor policy attack in RL brings additional attack opportunities in that the adversary can plan the attack sequentially: the immediate reward of a triggered round can even be good, as long as its state transition leads to much worst long-term rewards. Initial empirical work has shown backdoor policy attack is a valid security concern [6][7][19], but there has been neither formal attack specification nor provable defense.

This paper makes the following contributions: 1. We formally define backdoor policies and associated trigger function in RL called "subspace trigger". 2. We present a defense algorithm with performance guarantees against all subspace trigger based adversaries. 3. We empirically verify the performance of our sanitization algorithm on two Atari game environments.

Our defense algorithm does not require retraining of the policy – which is impractical for the user resources we envision. Instead, it is a wrapper method around the backdoor policy, rendering it insensitive to triggers and hence harmless. For this reason we call our defense "sanitization". The key idea is to project potentially triggered states to a safe subspace. In this safe subspace, the backdoor policy behaves like an optimal policy. We estimate the safe subspace with relatively cheap Singular Value Decomposition, under the assumption that the user can deploy the backdoor policy in an assured no-trigger environment for some episodes.

## 2   Related Work

The vast majority of backdoor attack literature targets supervised learning. Some work requires original training data and expensive model retraining [16] [11] [8] [12] [3] [10] [1] which we avoid. Some require surgical modification to the whitebox backdoor model [20] [4] [9] while our method is a wrapper. We also do not attempt to reverse engineer the trigger as done in [18].

Our method is in spirit closely related to [12], which uses autoencoder to find the equivalent to our "safe subspace". The connection is not incidental: our SVD is a special case of autoencoder. Nonetheless, we study backdoor policies in RL, where the key difference is that damage is inflicted via long term return instead of immediate reward while their work is in supervised learning setting and is empirical.

The study on backdoor policies in RL is only recently emerging [7][6][19]. Most of these work are again empirical, while we provide formal guarantees. Some of the recent works have also studied backdoor attacks in multi-agent RL setting - [19] proposed a backdoor attack where the backdoor behavior is triggered through a specific sequence of actions by the opponent which is clearly different from our setting where the attacker injects trigger directly in the state space. A follow-up work [5] proposed an empirical strategy to detect backdoor action sequence triggers in this setting which is also different from ours.

## 3   A Formal Definition of Backdoor Policies in RL

Let $\mathcal{M} = (\mathcal{S}, \mathcal{A}, P, R, \mu, \gamma)$ denote the environment MDP with a continuous state space $\mathcal{S} = \mathbb{R}^D$, discrete action space $\mathcal{A}$, a transition function $P : \mathcal{S} \times \mathcal{A} \to \Delta(\mathcal{S})$, a reward function $R : \mathcal{S} \times \mathcal{A} \to [0, 1]$, an initial state distribution $\mu$ and a discounting factor $\gamma \in [0, 1)$. The objective of the agent is to find a policy that achieves the maximum value under the given MDP $\mathcal{M}$,

$$V_{\mathcal{M}}^* = \max_\pi V_{\mathcal{M}}^\pi \tag{1}$$

where $V_{\mathcal{M}}^\pi = \mathbb{E}\left[\sum_{t=0}^\infty \gamma^t R(s_t, \pi(s_t))\right]$ is the expected discounted cumulative rewards of following policy $\pi$ under MDP $\mathcal{M}$.

The attacker first chooses an optimal policy $\pi^*$ with a discounted state occupancy under $\mathcal{M}$ given as $d_{\mathcal{M}}^{\pi^*}(s) = (1 - \gamma) \sum_{t=0}^\infty \gamma^t \mathbb{P}(s_t = s | s_0 \sim \mu, \pi^*)$. Let $T \subset \mathcal{S}$ denote the support of $d_{\mathcal{M}}^{\pi^*}$, i.e. $T$ is the smallest closed subset of $\mathbb{R}^D$ s.t. $\mathbb{P}(T) = 1$. For simplicity, we assume that $\mathbb{E}_{s \sim d_{\mathcal{M}}^{\pi^*}}[s] = 0$. This assumption keeps the analysis clean but our algorithm and its results can be directly extended to the non-zero mean case. Now, consider the eigen decomposition of the state covariance matrix $\Sigma$ under

$d_{\mathcal{M}}^{\pi^*}$,

$$\Sigma = \mathbb{E}_{s \sim d_{\mathcal{M}}^{\pi^*}}[ss^\top] = \sum_{i=1}^{D} \lambda_i u_i u_i^\top \quad \text{where } \lambda_1 \geq \cdots \lambda_d \geq \lambda_{d+1} \geq \cdots \geq \lambda_D. \tag{2}$$

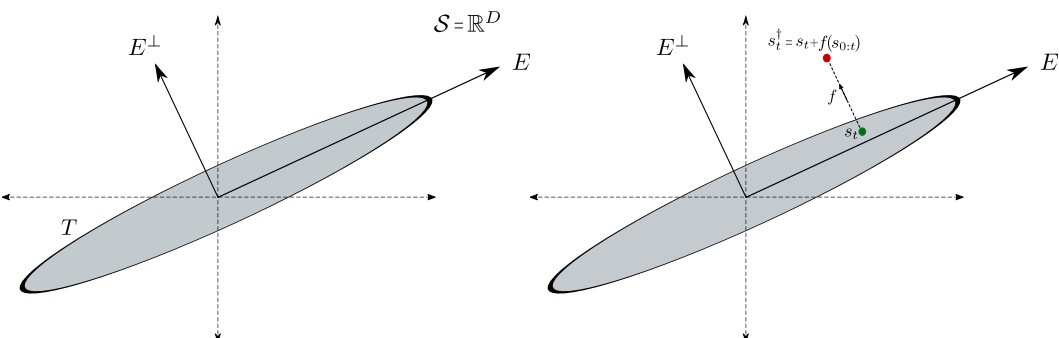

(a) An example of state space : the shaded region $T$ is the support of $d_{\mathcal{M}}^{\pi^*}$ and $E^\perp$ is the smallest $D - d$ dimensional eigen subspace of $\Sigma$.

(b) The adversary triggers a clean state $s_t$(green) in $E^\perp$ direction so that the agent observes the triggered state $s_t^\dagger$(red) and chooses a potentially malicious action $\pi^\dagger(s_t^\dagger)$ at $s_t$.

Figure 1: An example state space with state triggering by the adversary.

We denote by $E = \text{span}(\{u_i\}|_{i=1}^d)$ the top $d$ eigen subspace of $\Sigma$ and $E^\perp = \text{span}(\{u_i\}|_{i=d+1}^D)$ is its complement. The projection operator into $E, E^\perp$ is given by $Proj_E = \sum_{i=1}^d u_i u_i^\top$, $Proj_E^\perp = \sum_{i=d+1}^D u_i u_i^\top$ respectively, see Figure 1. We call $E$ the **safe subspace** of the state space (to be explained in Assumption 3).

**Assumption 1** *The occupancy distribution $d_{\mathcal{M}}^{\pi^*}$ has a bounded support along the smallest $D - d$ eigen-subspace $E^\perp$ i.e. $\exists C_0 \in \mathbb{R}$ s.t. under $d_{\mathcal{M}}^{\pi^*}$, $\mathbb{P}(\{s \in \mathcal{S} : \|Proj_{E^\perp}(s)\|_2 \leq C_0\}) = 1$. Further, we assume that $d_{\mathcal{M}}^{\pi^*}$ is $K$-subGaussian.*

In many real world examples, states are often bounded along standard basis directions so this assumption is reasonable. It gives the attacker the flexibility to insert into $\pi^\dagger$ arbitrary backdoor behaviors on states outside the bounded support $T$ while preserving the performance of optimal policy inside $T$.

**Backdoor Policy** $\pi^\dagger$    The attacker then constructs a backdoor policy $\pi^\dagger$, which behaves the same as $\pi^*$ everywhere in the support $T$ of $d_{\mathcal{M}}^{\pi^*}$, but outside this support $\pi^\dagger$ can differ arbitrarily. However, we assume that overall $\pi^\dagger$ is $L$-Lipschitz.

**Assumption 2** *The backdoor policy $\pi^\dagger$ is $L$-Lipschitz smooth i.e.*

$$\exists L \in \mathbb{R} \quad s.t. \quad \forall s, s' \in \mathcal{S}, \|\pi^\dagger(s) - \pi^\dagger(s')\|_1 \leq L\|s - s'\|_2.$$

In practice, people often use gradient descent trained neural network policies and it is reasonable to assume them to be Lipschitz smooth [2] [14]. By letting $\pi^\dagger = \pi^*$ on $T$, the support of $d_{\mathcal{M}}^{\pi^*}$, the attacker remains stealthy in a clean (non-triggered) environment and can coax the agent into believing that the backdoor policy $\pi^\dagger$ is a good one.

**Subspace Trigger function** $f$    In tandem with the backdoor policy $\pi^\dagger$, the attacker also prepares a trigger function $f : (\mathcal{S})^* \to E^\perp$ to be used during deployment, where $(\mathcal{S})^*$ is the set of all possible state sequence in the history. Importantly, such a trigger function restricts the adversary to only inject triggers in $E^\perp$ subspace and hence we call it "subspace trigger" function. Also, note that $f$ is adaptive in that it can depend on the entire history. For simplicity, we consider $f$ that only depends on history of states but it can be trivially generalized to include history of actions and rewards as well.

At any time step $t$ during deployment where the agent uses $\pi^\dagger$, the adversary can inject a trigger $f(s_{0:t})$ ("the sticker") into the current state $s_t$ so that the agent perceives a triggered state $s_t^\dagger = s_t + f(s_{0:t})$. Consequently, the agent will take a potentially harmful action $a_t^\dagger \sim \pi^\dagger(s_t^\dagger)$ at state $s_t$. We emphasize that the trigger only applies to the agent's perceived state, i.e. the underlying MDP state is still at $s_t$. As interaction protocol 1 specifies, transition and reward happen according to $(s_t, a_t^\dagger)$, not $(s_t^\dagger, a_t^\dagger)$. By carefully designing $f$ and coupling the behavior of backdoor policy on the triggered states, the adversary can either drastically decrease the value the agent receives, or force the agent to perform a nefarious action sequence.

Given an agent using a backdoor policy $\pi^\dagger$, and an adversary with a trigger function $f$, the interaction between them in an environment (possibly backdoored) takes place as follows:

---

**Procedure 1** Interaction Protocol in a Backdoor Environment

---

**Entities:** MDP environment $\mathcal{M}$, RL agent accepted backdoor policy $\pi^\dagger$ from the adversary, adversary has trigger function $f$.

1: the environment draws $s_0 \sim \mu$.
2: **for** $t = 0, 1, \cdots$ **do**
3:    the adversary injects trigger $f(s_{0:t})$ to state $s_t$ (adversary can choose to set $f(\cdot) = 0$ meaning no trigger).
4:    the agent observes $s_t^\dagger = s_t + f(s_{0:t})$ and takes the action $a_t^\dagger \sim \pi^\dagger(s_t^\dagger)$.
5:    the environment evolves $s_{t+1} \sim P(\cdot|s_t, a_t^\dagger)$, $r_t \sim R(s_t, a_t^\dagger)$

---

**Assumption 3** *The trigger function $f : (\mathcal{S})^* \to E^\perp$ is adaptive and the adversary can only inject a trigger in the $E^\perp$ subspace of the state space $\mathcal{S}$. Further, we assume that the perceived triggered states are $B$-bounded in expectation as below,*

$$\forall \pi, \forall t \in \mathbb{N} \quad \mathbb{E}_{s_{0:t} \sim d_{\mathcal{M}}^{\pi \circ f, 0:t}} \left[ \|(s_t + f(s_{0:t}))\|_2 \right] \leq B$$

*where $d_{\mathcal{M}}^{\pi \circ f, 0:t}$ denotes the distribution of partial state trajectory up to time step $t$ under policy $\pi$ and trigger function $f$.*

This assumption requires the adversary to keep the perceived triggered states to be $B$-bounded in expectation at every time step. Note that the adversary can still sometime inject large triggers as long as it keeps the average triggered states bounded.

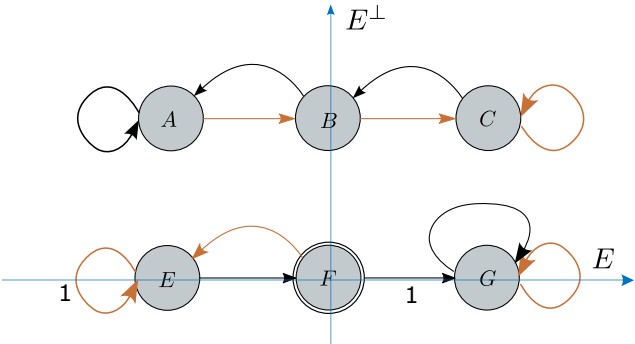

Figure 2: An example MDP under subspace trigger attack.

As an example, we present a toy MDP in Figure 2 where states are in $\mathbb{R}^2$. The initial state is always state $F$. There are two actions: left and right, and non-zero rewards are mentioned below an action. Transition is deterministic. Clearly, an optimal policy $\pi^*$ is to take the left action at $F$ and $E$ which yields a zero immediate reward but $\gamma/(1-\gamma)$ long term return. The x-axis is the safe subspace $E$. Note that the states $A$ to $C$ are non-reachable in the clean environment which gives the attacker leverage to hide the backdoor actions. The attacker's backdoor policy $\pi^\dagger$ is denoted by the orange

arrows in the above diagram. In particular, $B$ is far enough from $F$ so that the deterministic choice $\pi^\dagger(B) =$ "right" is allowed by the Lipschitz condition.

In the backdoor environment, when the agent starts at state $s_0 = F$, the adversary injects a trigger $f(F) \in \mathbb{R}^2$ in upward direction along $E^\perp$ so that the agent perceives its state as $s_0^\dagger = F + f(F) = B$ instead. The agent will then choose action $a_0^\dagger =$ "right" as directed by its policy $\pi^\dagger(B)$. However, the underlying MDP has its true state at $F$, so the agent moves to $G$. The agent may be happy that it received an immediate reward of 1 compared to 0 reward on left. However, the readers can see the agent is now doomed in the long term without further attacks. For $\gamma$ close to 1, the optimal value $\frac{\gamma}{1-\gamma}$ in the clean environment can be arbitrarily large while the value under the above attack is 1. This example contrasts sharply with backdoor attacks on supervised learning which focuses on instantaneous gratification.

Given a policy $\pi$ and a trigger function $f$, the value of the policy $\pi$ in the triggered environment is given as,

$$V_{\mathcal{M},f}^\pi = \mathbb{E}\left[\sum_{t=0}^\infty \gamma^t R(s_t, \pi(s_t + f(s_{0:t})))\right] = \mathbb{E}\left[\sum_{t=0}^\infty \gamma^t R(s_t, \pi \circ f(s_{0:t}))\right] = V_{\mathcal{M}}^{\pi \circ f}$$

Here, we note that the trigger function $f$ affects the value of the agent only through action selection. Also, since $f$ is adaptive (i.e. it depends on current state and history), the composition of a Markovian policy $\pi$ with the trigger function $f$ leads to a non-Markovian policy $\pi \circ f$.

**Goal of the defender**    The defender is provided with a backdoor policy $\pi^\dagger$, and has a sample of interactions between $\pi^\dagger$ and the clean environment $\mathcal{M}$. This is realistic in many applications: for example, even if a user does not trust the driving policy $\pi^\dagger$ she downloads, she can test drive the car for a few days in an enclosed driving facility. The goal of the defender is to sanitize the backdoor policy $\pi^\dagger$ such that the sanitized policy performs near-optimally even in the presence of trigger function $f$.

## 4   Sanitization Algorithm and Guarantees

We propose Algorithm 2 to sanitize and render backdoor policy harmless. Our sanitization algorithm works in an unsupervised manner by first recovering an estimate of the safe subspace using the clean samples from $d_{\mathcal{M}}^{\pi^\dagger}$ and projecting every states onto this empirical clean subspace to sanitize the state before feeding into the backdoor policy $\pi^\dagger$ (recall the agent is stuck with $\pi^\dagger$ since she does not have the resources to retrain).

---

**Algorithm 2** Defense through subspace sanitization

---

**Entities:** MDP environment $\mathcal{M}$, RL agent with policy $\pi^\dagger$, adversary with trigger function $f$.
**Inputs:** sample access to clean environment $\mathcal{M}$, number of clean samples $n$.

1: **Sanitization phase :**
2:     run $\pi^\dagger$ for $n$ clean episodes, collect $\{s_j\}|_{j=1}^n \overset{i.i.d}{\sim} d_{\mathcal{M}}^{\pi^\dagger}$.
3:     calculate the empirical covariance $\Sigma_n = \frac{1}{n}\sum_{j=1}^n s_j s_j^\top$ and its eigen decomposition,

$$\Sigma_n = \sum_{i=1}^D \hat{\lambda}_i \hat{u}_i \hat{u}_i^\top \quad \text{where } \hat{\lambda}_1 \geq \cdots \hat{\lambda}_d \geq \hat{\lambda}_{d+1} \geq \cdots \geq \hat{\lambda}_D.$$

4:     construct the empirical projection operator $Proj_{E_n} = \sum_{i=1}^d \hat{u}_i \hat{u}_i^\top$.
5:     define the sanitized policy $\pi_{E_n}^\dagger : \mathcal{S} \to \Delta(A)$ s.t. $\forall s \in \mathcal{S}$, $\pi_{E_n}^\dagger(s) = \pi^\dagger \circ Proj_{E_n}(s)$.
6: **Deployment phase :**
7:     at every time step $t$, the agents takes the action $\pi_{E_n}^\dagger(s_t^\dagger)$.

---

Algorithm 2 has a strong guarantee. We recall that the backdoor policy performs optimally in the clean environment i.e. $V_{\mathcal{M}}^{\pi^\dagger} = V_{\mathcal{M}}^*$. So, we would like the performance of sanitized policy $\pi_{E_n}^\dagger$ in the triggered environment$(\mathcal{M}, f)$ to be as close to the performance of the optimal policy in the clean environment. Thus, we are interested in upper bounding the performance difference $V_{\mathcal{M}}^* - V_{\mathcal{M}}^{\pi_{E_n}^\dagger \circ f}$.

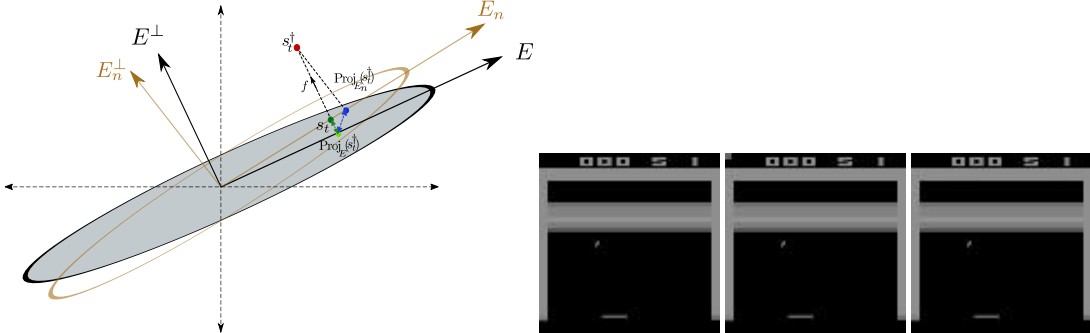

(a) Pictorial representation of sanitization process.     (b) Clean state.    (c) Triggered state. (d) Sanitized state.

Figure 3: On the left, a clean state $s_t$ is triggered into $s_t^\dagger$ which is projected back onto the estimated safe subspace $E_n$ as $\mathrm{Proj}_{E_n}(s_t^\dagger)$ during sanitization. The agent finally takes the action $\pi^\dagger(\mathrm{Proj}_{E_n}(s_t^\dagger))$ instead of $\pi^\dagger(s_t^\dagger)$. Figure (b,c,d) on the right, instantiates this process on the breakout game example. Note that the square trigger (top left) in the triggered state in figure (c) is filtered out after projection onto the empirical safe subspace as in figure (d).

**Theorem 4** *Let $\pi^\dagger$ be the backdoor policy, $f$ be the triggered function and let $d_{\mathcal{M}}^{\pi^\dagger}$ denote the discounted state occupancy distribution of $\pi^\dagger$ under clean environment MDP $\mathcal{M}$. Further, let $\delta_* = \lambda_d - \lambda_{d+1} > 0$ be the eigen gap between the safe subspace $E$ and $E^\perp$ as defined in (2). Under assumptions 1, 2, 3 stated above, for a defender that uses the sanitized policy $\pi_{E_n}^\dagger$ as defined in Algorithm 2 against an attacker with trigger function $f$, we have that,*

$$\forall \epsilon > 0, \forall \delta > 0, \ if \ n \geq \frac{CdB^2L^2K^4\|\Sigma\|_2^2}{\delta_*^2(1-\gamma)^4 \cdot \epsilon^2}\left(D + \log\frac{2}{\delta}\right), \ with \ probability \ \geq 1-\delta, \qquad (3)$$

$$V_{\mathcal{M}}^* - V_{\mathcal{M}}^{\pi_{E_n}^\dagger \circ f} \leq \underbrace{\frac{L}{(1-\gamma)^2}\cdot\sqrt{\sum_{i=d+1}^{D}\lambda_i}}_{approximation\ error} + \underbrace{\epsilon}_{estimation\ error} \qquad (4)$$

*where $L$ is the Lipschitz constant of the policy $\pi^\dagger$ in assumption 2, $B$ is the upper bound on the expected observed triggered states in assumption 3, $K$ is the sub-Gaussian parameter of the state-occupancy distribution $d_{\mathcal{M}}^{\pi^\dagger}$, $\gamma$ is the discount factor and $C$ is some fixed constant.*

We make the following remarks:

- Our safety guarantee is an $(\epsilon, \delta)$-PAC style guarantee with both approximation and estimation error terms. The first term is a fixed approximation error $\epsilon_{app}$ that the defense algorithm has to suffer even in the presence of infinite clean samples. However, if the eigen energy of $E^\perp$ subspace $\sum_{i=d+1}^{D}\lambda_i = 0$, i.e. if the adversary injects trigger only in the null eigen spaces of $\Sigma$, the defender can avoid the approximation error.
- The second term $\epsilon$ is the estimation error which scales as $O(1/\sqrt{n})$ and can be made as small as required with more clean samples.
- The overall sample complexity grows polynomially in the effective horizon $1/(1-\gamma)$, the inverse eigen-gap $\frac{1}{\delta_*}$, the inverse estimation error $\frac{1}{\epsilon}$ and the dimension of state space $D$. We note that the lower the eigen separation between the safe subspace and its complement, the more difficult will it be for the defender to recover the safe subspace using samples; hence more difficult the defense.

**Proof:** The key step in the proof is the following value decomposition:

$$V_{\mathcal{M}}^* - V_{\mathcal{M}}^{\pi_{E_n}^\dagger \circ f} = V_{\mathcal{M}}^{\pi^\dagger} - V_{\mathcal{M}}^{\pi_{E_n}^\dagger \circ f} \qquad (5)$$

$$= \underbrace{V_{\mathcal{M}}^{\pi^\dagger} - V_{\mathcal{M}}^{\pi_E^\dagger}}_{(1)} + \underbrace{V_{\mathcal{M}}^{\pi_E^\dagger} - V_{\mathcal{M}}^{\pi_E^\dagger \circ f}}_{(2)} + \underbrace{V_{\mathcal{M}}^{\pi_E^\dagger \circ f} - V_{\mathcal{M}}^{\pi_{E_n}^\dagger \circ f}}_{(3)} \qquad (6)$$

Term (1) leads to an approximation error in the clean environment which cannot be avoided even in presence of infinite samples. Term (2) is the performance difference of the true projected policy $\pi_E^\dagger = \pi^\dagger \circ Proj_E$ when acting on clean vs triggered environment. Term (3) is the estimation error that arises due to the fact that the defender only gets sample access to the clean environment.

Since the triggers always lie in $E^\perp$ (by assumption 3), projection of $s_t^\dagger$ onto $E$ engulfs the trigger part of it effectively reducing the second error term to zero. To see this, simply observe that $\pi_E^\dagger(s_t + f(s_{0:t})) = \pi^\dagger(Proj_E(s_t + f(s_{0:t}))) = \pi^\dagger(Proj_E(s_t)) = \pi_E^\dagger(s_t)$ using linearity of projection operator and orthogonality of triggers $f(\cdot)$ to $E$ subspace, which implies,

$$V_{\mathcal{M}}^{\pi_E^\dagger \circ f} = \mathbb{E}\left[\sum_{t=0}^\infty \gamma^t R(s_t, \pi_E^\dagger(s_t + f(s_{0:t})))\right] = \mathbb{E}\left[\sum_{t=0}^\infty \gamma^t R(s_t, \pi_E^\dagger(s_t))\right] = V_{\mathcal{M}}^{\pi_E^\dagger}. \qquad (7)$$

We now bound the approximation error in the clean environment, the term (1) in (6):

$$
\begin{aligned}
V_{\mathcal{M}}^{\pi^\dagger} - V_{\mathcal{M}}^{\pi_E^\dagger} &\overset{(1)}{=} \frac{1}{1-\gamma}\mathbb{E}_{s\sim d_{\mathcal{M}}^{\pi^\dagger}}\left[\left(\pi^\dagger(s) - \pi_E^\dagger(s)\right)^\top Q^{\pi_E^\dagger}(s,\cdot)\right] \\
&\overset{(2)}{\le} \frac{1}{1-\gamma}\mathbb{E}_{s\sim d_{\mathcal{M}}^{\pi^\dagger}}\left[\|\pi^\dagger(s) - \pi_E^\dagger(s)\|_1 \|Q^{\pi_E^\dagger}(s,\cdot)\|_\infty\right] \\
&\overset{(3)}{\le} \frac{L}{1-\gamma}\cdot \|Q^{\pi_E^\dagger}\|_\infty \cdot \sqrt{\mathbb{E}_{s\sim d_{\mathcal{M}}^{\pi^\dagger}}\|(I - Proj_E)s\|_2^2} \\
&\overset{(4)}{\le} \frac{L}{(1-\gamma)^2}\cdot \sqrt{\mathbb{E}_{s\sim d_{\mathcal{M}}^{\pi^\dagger}}\|\sum_{i=d+1}^D u_i u_i^\top s\|_2^2} \\
&\overset{(5)}{=} \frac{L}{(1-\gamma)^2}\cdot \sqrt{\sum_{i=d+1}^D \lambda_i}
\end{aligned}
\qquad (8)
$$

where (1) uses performance difference lemma (Lemma 8), (2) is Holders inequality, (3) uses $L$-Lipschitzness of $\pi^\dagger$ and Jensen's inequality, (4) uses $\|Q^{\pi_E^\dagger}\|_\infty \le 1/(1-\gamma)$ and (5) follows from the following :

$$
\begin{aligned}
\mathbb{E}_{s\sim d_{\mathcal{M}}^{\pi^\dagger}}\|\sum_i u_i u_i^\top s\|_2^2 &= \sum_i \mathbb{E}_{s\sim d_{\mathcal{M}}^{\pi^\dagger}}\left[u_i^\top ss^\top u_i\right] = \sum_i \mathbb{E}_{s\sim d_{\mathcal{M}}^{\pi^\dagger}}\left[\text{tr}(ss^\top u_i u_i^\top)\right] \\
&= \sum_i \text{tr}(\mathbb{E}_{s\sim d_{\mathcal{M}}^{\pi^\dagger}}\left[ss^\top\right] u_i u_i^\top) = \sum_i \text{tr}(\lambda_i u_i u_i^\top) = \sum_i \lambda_i. \qquad (9)
\end{aligned}
$$

Next, we bound the estimation error in the triggered environment, the third term in (6). Let $\pi = \pi^\dagger \circ Proj_E$ and $\pi' = \pi^\dagger \circ Proj_{E_n}$, then the difference in value of these policies in a triggered environment is given as

$$V_{\mathcal{M}}^{\pi \circ f}(s_0) - V_{\mathcal{M}}^{\pi' \circ f}(s_0) \qquad (10)$$

$$
\begin{aligned}
&\overset{(1)}{=} \sum_{t=0}^\infty \gamma^t \mathbb{E}_{s_{0:t}\sim d_{\mathcal{M}}^{\pi\circ f,0:t}}\left[Q_{\mathcal{M}}^{\pi'\circ f}(s_{0:t}, \pi \circ f(s_{0:t})) - Q_{\mathcal{M}}^{\pi'\circ f}(s_{0:t}, \pi' \circ f(s_{0:t}))\right] \\
&\overset{(2)}{\le} \sum_{t=0}^\infty \gamma^t \mathbb{E}_{s_{0:t}\sim d_{\mathcal{M}}^{\pi\circ f,0:t}}\left[\|\pi \circ f(s_{0:t}) - \pi' \circ f(s_{0:t})\|_1 \|Q_{\mathcal{M}}^{\pi'\circ f}(s_{0:t}, \cdot)\|_\infty\right] \\
&\overset{(3)}{\le} \frac{1}{1-\gamma}\sum_{t=0}^\infty \gamma^t \mathbb{E}_{s_{0:t}\sim d_{\mathcal{M}}^{\pi\circ f,0:t}}\left[\|(\pi^\dagger(Proj_E(s_t + f(s_{0:t}))) - \pi^\dagger(Proj_{E_n}(s_t + f(s_{0:t}))))\|_1\right] \\
&\overset{(4)}{\le} \frac{1}{1-\gamma}\sum_{t=0}^\infty \gamma^t L \mathbb{E}_{s_{0:t}\sim d_{\mathcal{M}}^{\pi\circ f,0:t}}\left[\|(Proj_E - Proj_{E_n})(s_t + f(s_{0:t}))\|_2\right] \\
&\overset{(5)}{\le} \frac{L}{1-\gamma}\cdot \|Proj_E - Proj_{E_n}\|_2 \sum_{t=0}^\infty \gamma^t \mathbb{E}_{s_{0:t}\sim d_{\mathcal{M}}^{\pi\circ f,0:t}}\left[\|(s_t + f(s_{0:t}))\|_2\right] \\
&\overset{(6)}{\le} \frac{B\cdot L}{(1-\gamma)^2}\|Proj_E - Proj_{E_n}\|_2 \qquad (11)
\end{aligned}
$$

where (1) follows from performance difference lemma for non-Markovian policies (Lemma 8), (2) follows from Holder's inequality, (3) follows from $\|Q\|_\infty \leq 1/(1-\gamma), (4)$ follows from $L$-Lipschitzness of $\pi^\dagger$, (5) follows from $L_2$ matrix vector norm inequality and (6) uses the $B$-boundedness of the trigger functions $f$ as defined in 3.

Now, if $n \geq \frac{CdB^2L^2K^4\|\Sigma\|^2}{\delta_*^{\,2}(1-\gamma)^4 \cdot \epsilon^2} \left(D + \log \frac{2}{\delta}\right)$, using Lemma 6 with (11), we can upper bound the estimation error by $\epsilon$, i.e.

$$V_{\mathcal{M}}^{\pi_E^\dagger \circ f} - V_{\mathcal{M}}^{\pi_{En}^\dagger \circ f} \leq \epsilon. \tag{12}$$

*Finishing the proof of Theorem 4*: We plug back (7), (8), (12) in (6) to conclude the proof. $\qquad \square$

**Computational Complexity**  The computation complexity of our subspace sanitization algorithm is $O(D^3 + T \cdot D^2)$ where $D$ is the dimension of the state space and $T$ is the total number of time step the agent runs during deployment. The algorithm incurs a one time cost of $O(D^3)$ for computing the SVD of the empirical covariance matrix and the projection operation at each time step of deployment requires $O(D^2)$ compute time.

# 5   Experiments

In this section, we present some experimental results that validate our sanitization algorithm against backdoor attacks in Atari game environments. We performed the sanitization experiments on backdoor policies in two Atari games: (i) Boxing-Ram game with a vector RAM state space, (ii) Breakout game with an image state space. In each case, we acted as the attacker and pre-trained a backdoor policy using a reward poisoning schema as described in [7]. The backdoor policy was trained to a level such that the agent performs well in clean environment but fares poorly in the triggered environment. In each case, we used our sanitization algorithm to sanitize the backdoor policy and reported their results.

## 5.1   Boxing-Ram game

**Environment and attack specification**  The state consists of a 32 dimensional byte RAM vector: a concatenation of four 8-byte vectors representing information stored in the simulator's RAM from the latest four time steps. The attacker exploits an unused byte (which is always zero in the clean environment) to inject triggers directly into the vector RAM state space. Concretely, $f = 255 \cdot e_{28}$ where $e_{28}$ is $28^{th}$ canonical basis vector in $\mathbb{R}^{32}$. The action space consists of 18 different actions: moving left, right, up, down, punch, and combinations thereof.

**Backdoor policy**  The backdoor policy consists of a 4 layered fully connected neural network with second layer shared to the critic network. The joint actor critic network is trained using PPO algorithm [15] implemented in Pytorch. The backdoor policy has been trained using an environment poisoning scheme following [6] to train the agent to seek low value in the triggered environment. The policy has been trained to a level such that it receives high returns in the clean environment. However, when the attacker activates the trigger, the agent will move close to the opponent to get beaten down without fighting back, thus minimizing its return.

## 5.2   Breakout game

**Environment and the attack specification**  We consider a version of breakout game where the states consist of a stack of four down-scaled board images ($4 \times 84 \times 84$) from the last 4 time steps and the action space consists of three actions: move left, move right, and do not move. The attack directly takes place in the image state space where the attacker injects a $6 \times 6$ pixel square trigger on the top left part of the arena, see Figure 3.

**Backdoor policy**  The backdoor policy is a neural network with two convolutional layers followed by two dense layers with ReLU activation units. The policy has been trained using environment poisoning attack scheme following [7] with a backdoor objective to force the agent to take 'do not move' action in the triggered states. Empirically, the policy has been trained to a level such that

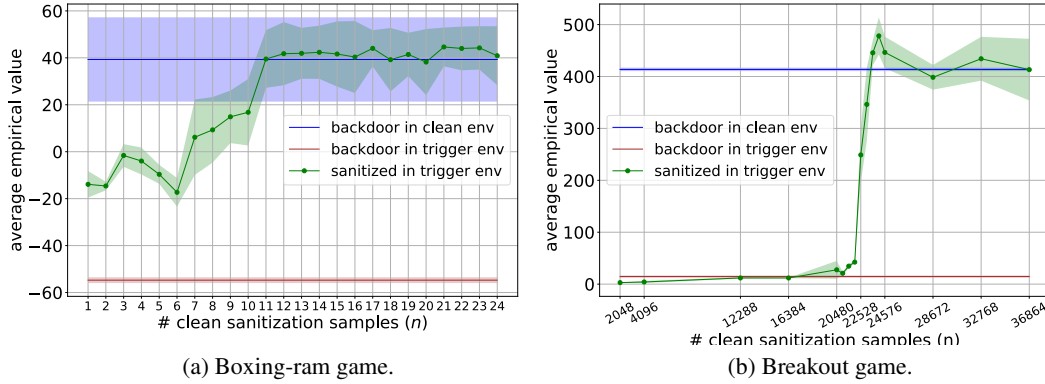

(a) Boxing-ram game.            (b) Breakout game.

Figure 4: Performance of the backdoor policy in the clean environment (blue), and in the triggered environment (brown). With sufficient clean sanitization samples ($n$), our sanitized policy recovers back the clean performance (green). Shading is $\pm$ standard deviation.

it consistently performs very well in the clean environment receiving high returns. However, in presence of trigger, it falters into not moving at all thus losing lives very quickly.

### 5.3 Empirical results of our sanitization algorithm

For a fixed sanitization sample count $n$, the defender collects $n$ clean episodes from the environment and chooses an independent sample from each episode to get $n$ samples $d_{\mathcal{M}}^{\pi^\dagger}$. Empirically, we observe that choosing roughly $2n$ correlated samples also works for sanitization in these examples. It then constructs a sanitized policy (as defined in Algorithm 2) using top $d$ eigen bases of the empirical covariance matrix. The safe space dimension $d$ is decided using the eigen gap in the empirical covariance matrix (all eigenvectors with eigenvalues $\geq 10^{-10}$ are chosen as 'safe subspace', see section 5.4 for further discussion).

In Figure 4, the blue and brown lines show the performance of backdoor policy in clean and triggered environment, respectively. The $y$-axis shows the mean and standard deviation of empirical agent value obtained from 4 independent trials. In each trial, we estimated the empirical value from averaging the returns obtained from 5 independent episodes sampled using the respective (policy, trigger) pair. We observe that the backdoor policy performed well in the clean environment (blue). However, its performance dropped significantly in the presence of triggers (brown). Next, the green line shows the performance of the sanitized policy constructed using $n$ clean samples (on $x$-axis) from $d_{\mathcal{M}}^{\pi^\dagger}$. We clearly observe that in both the examples, the performance of sanitized policy increases with $n$ and after a sufficient number of clean samples the sanitized policy recovers backs the original clean performance (of backdoor policy in the clean environment, blue line) even when acting in the triggered environment. This empirically verifies the success of our sanitization algorithm against subspace backdoor attacks.

### 5.4 Dependence of our algorithm on the dimension $d$ of safe subspace

Our theory assumes that the defender knows the dimension of the safe space $d$ which might not always hold true in practice. Though, the defender still has an access to eigen spectrum of the clean empirical covariance matrix $\Sigma_n$ which it can use to estimate $d$ and the corresponding 'safe subspace'. A good estimate of $d$ is important, because both underestimation and overestimation can lead to loss in performance. Specifically, with underestimation the defender might lose important dimensions in the safe subspace, and with overestimation the defender may include spurious dimensions from $E^\perp$ hampering effective sanitization in both scenarios. This phenomenon is depicted by the empirical value curve (green) in Figure 5, where we see that the performance of the sanitized policy decreases both with underestimation and overestimation of $d$. In practice, one can choose $d$ based on the spectral gap of the empirical covariance matrix $\Sigma_n$. From the singular value spectrum curve (blue) in

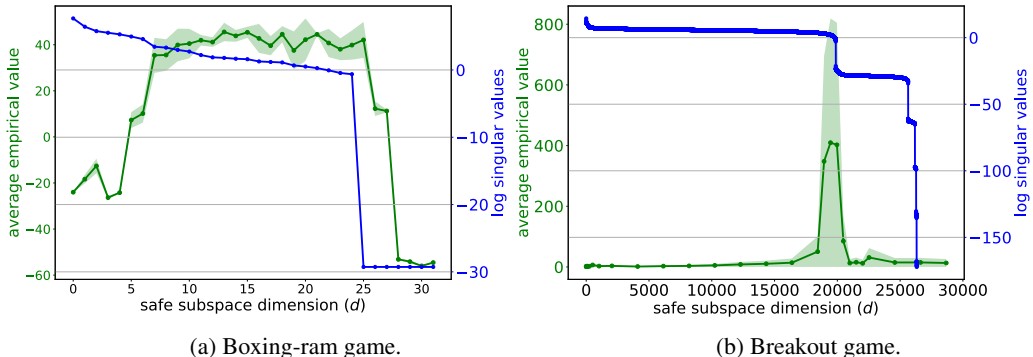

(a) Boxing-ram game.  (b) Breakout game.

Figure 5: Performance of sanitized policies with different safe space dimension $d$ (in green); Plot of log singular value spectrum of empirical covariance matrix (in blue) with $n = 40$ for boxing-ram game and $n = 32768$ for the breakout game.

Figure 5, we observe that the correct threshold corresponds to dimension just before the first major dip in singular values; which occurs after dimension 24 in Boxing-Ram game and near dimension 20000 in Breakout game. We used these thresholds to select $d$ in our experiments.

## 6 Conclusion

We formally specified backdoor policy with subspace trigger attacks in RL. We then proposed a sanitization algorithm that allows a user to safely use a backdoor policy under subspace trigger attackers. Our defense has the advantage of being a wrapper method and does not require expensive policy retraining. Our sanitization work makes RL safer and contributes to trustworthy AI. Future work will address the limitations of our defense, namely the need for clean environment interactions and the assumption that triggers reside in a subspace $E^{\perp}$.

**Acknowledgment** This work was supported in part by NSF grants 1545481, 1704117, 1836978, 2023239, 2041428, 2202457, ARO MURI W911NF2110317, and AF CoE FA9550-18-1-0166.

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
