# OpenReview forum: "Provable Defense against Backdoor Policies in Reinforcement Learning"
_NeurIPS.cc/2022/Conference — NeurIPS 2022 Accept_

### Official Review · Reviewer_5omh · 2022-07-10

**Rating:** 6
**Confidence:** 3
**Soundness:** 3 good
**Presentation:** 3 good
**Contribution:** 2 fair

**Summary:**

This paper formulate backdoor policy attack. It specifies a kind of backdoor policy attack and provides an algorithm to defense such attack. It provide a proof to bound the suboptimality of the policy sanitized by the defense. It also conducts experiments to verify their algorithm.

**Questions:**

1. The trigger function used in the experiment seems really simple, since the basis of the image of the trigger function is e_i, which greatly reduce the learning difficulty.


**Limitations:**

Their formulation only adapts to the case that the support of the image the trigger function is a linear subspace of $R^d$, which might be restricted for real-life application.

**Strengths And Weaknesses:**

Strength: Backdoor attack is an interesting problem, and the formulation in this paper is useful for theoretical study and algorithm design. They also provide an algorithm and prove the soundness of the algorithm both theoretical and empirixal.

Weakness: The writing of proof for the main theorem can be improved. Moreover, their algorithm and analysis can only  be used on the simple backdoor attack that consist with their assumption.

---

> ### Author Response · Authors · 2022-08-02
> **Response to Official Review of Paper12310 by Reviewer 5omh**
>
> We thank the reviewer for their valuable feedback. Please find our response below.
>
> Q1. The trigger function used in the experiment seems really simple, since the basis of the image of the trigger function is $e_i$, which greatly reduce the learning difficulty.
> - We note that square patch based triggers represent natural candidates that are likely to occur in practice and have been used in prior works like TrojDRL [1]. As far as learning difficulty is concerned, the complexity of learning the eigen subspaces is $O(D^3)$ irrespective of form the eigen vectors take. Thus, we think that the example with canonical basis vectors does not greatly reduce the learning difficulty.
>
> References :
>
> [1]. TrojDRL: Evaluation of Backdoor Attacks on Deep Reinforcement Learning, Panagiota Kiourti and Kacper Wardega and Susmit Jha and Wenchao Li, DAC 2020.

---

### Official Review · Reviewer_iqtz · 2022-07-10

**Rating:** 5
**Confidence:** 4
**Soundness:** 3 good
**Presentation:** 3 good
**Contribution:** 2 fair

**Summary:**

This paper proposes a defense method to deal with the backdoor attack on reinforcement learning (RL). Instead of re-training the RL policy or directly modifying the model parameters, the proposed method serves as a wrapper that projects the observation onto an estimated safe subspace and thus, eliminates the potential triggers. This helps the RL agent retain its performance even in triggered environments. The authors also provide theoretical analysis for their method.

**Questions:**

1. Strong assumption: this paper assumes that the adversary can only inject a trigger in the $E^{\bot}$ subspace of the state space and the proposed sanitization algorithm heavily relies on this assumption. In my personal opinion, this assumption is strong because the $E^{\bot}$ subspace being a convenient place to inject the trigger does not mean the adversary has to do that. Actually, even the adversary injects the trigger in the clean support, it may not affect the performance when the trigger is not present. We can use the Figure 2 as an example. When no trigger is present, the agent will turn left to state E, which is the optimal policy. Then if the adversary injects a trigger in state F and makes the agent think it is in state G (still in clean support), the agent will choose the right direction, which destroys the agent's performance. In this case, the sanitization algorithm may not work because state G is not in the $E^{\bot}$ subspace. In practical, I think it is very hard for the adversary to always inject a trigger in the $E^{\bot}$ subspace and there is also no incentive for them to do that. The adversary only cares whether the backdoor is insert into the victim model or not. For some domains, the successful backdoor may happen to have the trigger patterns that locate in the $E^{\bot}$ subspace. For other domains, the trigger patterns locating in the clean support may already be good enough. Therefore, the proposed method seems to only work for the cases where the triggers are only injected in the $E^{\bot}$ subspace. In addition, I am not sure whether the assumption 2 is practical or not. It seems to be an assumption solely for the theoretical analysis. It would be better if the authors can provide some intuitive explanations/examples to justify it.

2. The comparison with existing works: the experiments do not compare the proposed method with existing works. I agree with the authors that the backdoor defense for RL is different from that for supervised learning because the damage for RL is inflicted via the long term return instead of immediate rewards. However, in the sanitization algorithm, the authors just collect a set of states being visited, perform SVD on the state set, and then directly use the estimated projection operator to sanitize the observation. In the whole process, I do not see the authors explicitly handle the effect of long term return. Therefore, I think it is worthy to compare with the existing works for supervised learning that also pre-process the observations before feeding them into the trained model. For example, the work [12] that uses autoencoder could be a good candidate.

**Limitations:**

Please check the Questions section.

**Strengths And Weaknesses:**

Strength:
1. Originality: the idea of this paper is novel and the proposed method is a promissing exploration for the defense mechanism against backdoor attack on RL.

2. Clarity: the paper is organized well and the paper is easy to follow.

Weaknesses:
1. Quality: the results of this paper are sound and well supported by the conducted experiments. However, the method seems to rely on a strong assumption (please see detailed discussion in Question section). Also, the method is only tested in two Atari game, which is kind of limited.

2. Significance: the experiments prove the method can work in some scenarios. However, there are no comparisons with any existing works. Therefore, it is unclear how much this method advances the state of the art.

---

> ### Author Response · Authors · 2022-08-02
> **Response to Official Review of Paper12310 by Reviewer iqtz**
>
> We thank the reviewer for their valuable feedback. Please find our response to the raised questions and comments below.
>
>
> Q1.a. In my personal opinion, this assumption is strong because the $E^\perp$ subspace being a convenient place to inject the trigger does not mean the adversary has to do that.
> - We would like to politely disagree that our assumption is strong. Various past works on backdoor attacks have used the trigger functions that satisfy our assumption. For example : TrojDRL [1] have used the pixel based triggers that lie entire in the null space of the state occupancy covariance matrix of the near-optimal policy. Our assumptions is a generalized version of such triggers, where we also allow the adversary to put triggers in eigen subspace with strictly positive but small eigen values. Another work on backdoor RL [7], have used a very similar subspace based trigger.
> - Conversely, if one doesn't put any restriction on the adversary - say if the adversary is allowed to inject any arbitrary trigger, it can make the true state look like any other state and force the agent to take a bad action and the defender would have no hope of defense in presence of such backdoor attacks - note that such attacks would be very unrealistic as well. The example with trigger in $E$ you have pointed out conveys the same point. So, we believe that our assumption is the right place to start the theoretical defense analysis and leave further relaxation of our assumptions - say allowing triggers in non-linear subspace and so on - to future works.
>
> Q1.b. I am not sure whether the assumption 2 is practical or not. It seems to be an assumption solely for the theoretical analysis.
> - In practice, it is reasonable to assume that neural networks are Lipschitz smooth, though it has been shown that tightest Lipschitz constant of neural networks are difficult to be estimated. There have been number of works in this direction [2], [3], [4], [5]. Some recent works in adversarial learning have also tried to explicitly regularize neural network weights so as to learn lipschitz smooth networks that make it robust against test time adversarial attack. Hence, Lipschitz policies do occur in practice.
>
> Q2.a. I agree with the authors that the backdoor defense for RL is different from that for supervised learning because the damage for RL is inflicted via the long term return instead of immediate rewards. I do not see the authors explicitly handle the effect of long term return.
> - The defender needs careful planning for successful attack in RL. But on the defense side, as long as the defender can thwart the attack at each time step, any eventual long term planning by the adversary would be thwarted. This is what is achieved by the projection step of our algorithm.
>
> Q2.b. I think it is worthy to compare with the existing works for supervised learning that also pre-process the observations before feeding them into the trained model.
> - A well trained autoencoder with a $d$-dimensional latent linear layer is equivalent to doing a PCA with $d$ principal components - in the sense that both recover the same subspace. So, autoencoder could well be used as a substitute for PCA. We note that unlike [6] which was an empirical work, our work also provides theoretical guarantee on the performance of the PCA projection based sanitization algorithm under reasonable assumption. As such, our theoretical result also provides an insight into when the autoencoder defense proposed by [6] would work in practice.
> - As far as we know, there has been no prior work on successful defense against backdoor attacks in the setting we studied, let alone with a theoretical analysis. The paper TrojRL[1] (where we borrowed our experimental candidate from) have reported to have tried several defense methods from supervised learning area with no empirical success.
>
>
> References :
>
> [1]. TrojDRL: Evaluation of Backdoor Attacks on Deep Reinforcement Learning, Panagiota Kiourti and Kacper Wardega and Susmit Jha and Wenchao Li, DAC 2020.
>
> [2]. Efficient and Accurate Estimation of Lipschitz Constants for Deep Neural Networks, {Fazlyab}, Mahyar and {Robey}, Alexander and {Hassani}, Hamed and {Morari}, Manfred and {Pappas}, George J., arXiv 1906.04893 .
>
> [3]. Lipschitz regularity of deep neural networks: analysis and efficient estimation, Virmaux, Aladin and Scaman, Kevin, Neurips 2018.
>
> [4]. Regularisation of Neural Networks by Enforcing Lipschitz Continuity, Gouk, Henry and Frank, Eibe and Pfahringer, Bernhard and Cree, Michael J., arXiv, 2018.
>
> [5]. Lipschitz constant estimation of Neural Networks via sparse polynomial optimization, Latorre, Fabian and Rolland, Paul and Cevher, Volkan, arxiv 2020.
>
> [6]. Neural Trojans, Liu, Yuntao and Xie, Yang and Srivastava, Ankur, arXiv 2017.
>
> [7]. The TrojAI Software Framework: An OpenSource tool for Embedding Trojans into Deep Learning Models, Karra, Kiran and Ashcraft, Chace and Fendley, Neil, arXiv 2020.

---

> > ### Comment · Reviewer_iqtz · 2022-08-06
> > **Thanks for the authors' response**
> >
> > The authors' response has cleared most of my concerns. But I still don't get why putting the trigger in eigen subspace with strictly positive and large eigen values would be an unrealistic attack. Is it because of the possibility of being detected? What kind of detection ability you have assumed for the defender?

---

> > > ### Author Response · Authors · 2022-08-08
> > > **Follow up reply to Reviewer iqtz**
> > >
> > > We are not saying that putting triggers in high variance directions would be unrealistic.  For example, another reasonable assumption could be to allow the adversary to inject large triggers in any directions under a true Gaussian occupancy measure and we think a projection based sanitization strategy would still work(perhaps with a different notion of projection like projecting the states onto a estimated heavy occupancy ellipsoid). It would indeed be very interesting to extend our framework and results to tackle such attacks in the future work. At this point, we would like to clarify some more points.
> > >
> > > - On one extreme end, there has been recent works(in supervised learning) [8] which provide an impossibility result on defense in unrestricted black box setting - essentially it states that under cryptographic hardness assumptions, if the attacker is allowed to train and hand over a black-box model to the user then a practical(polynomial time) defense algorithm is impossible. We think that result easily extends to the RL setting as well and so it will be impossible to guarantee a secure system without certain reasonable restrictive assumptions on the abilities of the attacker. Note that their assumption that the attacker can hand over a backdoor black-box model to the user and arbitrarily perturb all the components in the signature part of the input at test time would be considered quite unrealistic.
> > > - On the other end, our work is an important stepping stone in the direction of provable defense as it not only captures and generalizes some of prior works [1] [7] in backdoor-RL(those that have indeed been shown to be an effective way to doing backdoor attack in RL) but also provides an effective handle to provably sanitize the backdoor model against such attacks.
> > >
> > > In conclusion, currently it is an open question on what is the right kind of restriction that a user should put on the third party so as to prevent a backdoor attack. And it would indeed be very interesting to study stronger but reasonable attack models under which the defender can provide provable defense guarantees in future works.
> > >
> > > Additional Reference :
> > > [8] Planting Undetectable Backdoors in Machine Learning Models, Goldwasser, Shafi and Kim, Michael P. and Vaikuntanathan, Vinod and Zamir, Or, [arXiv 2022](https://arxiv.org/abs/2204.06974).

---

> > > > ### Comment · Reviewer_iqtz · 2022-08-09
> > > > **Thanks for the reply**
> > > >
> > > > I have read the authors' reply and will raise the score to 5. I suggest the authors emphasize the restriction on the attacker ability more clearly to avoid any over-claims about the capability of the defense method.

---

### Official Review · Reviewer_uJFB · 2022-07-10

**Rating:** 6
**Confidence:** 3
**Soundness:** 3 good
**Presentation:** 4 excellent
**Contribution:** 3 good

**Summary:**

This paper proposes a defense algorithm against backdoor attacks for reinforcement learning. Specifically, the defense is based on Singular Value Decomposition of the covariance matrix of state occupancy. The algorithm sanitizes the poisoned policy by projecting states into a ``safe subspace'' formed by a low-rank decomposition of the covariance matrix. Based on several assumptions on the state occupancy structure, the authors show that the proposed algorithm guarantees a bounded different between the optimal policy's value and the sanitized policy's value. Experiments on a vector-input game and a pixel-input game verify the effectiveness of the proposed algorithm.

**Questions:**

1. What is the $K$ in the sample complexity result?

2. What is the computational complexity of the algorithm in theory and in experiments?

**Limitations:**

The paper briefly discusses the limitations of the work in Section 6.

**Strengths And Weaknesses:**

Strengths:

The tackled problem, defending against backdoor policy attack, is important. The proposed algorithm applies SVD to filter out triggers, which makes intuitive sense.

The experiment results are good. It is surprising that the proposed SVD-based defense also works for pixel inputs.

The presentation of this paper is good, and easy to follow. The authors make the attack and defense problem clear, with nice visualization figures. The mathematical notations are also clean to me.

---

Weaknesses:

The algorithm, although makes intuitive sense, may have some difficulties of scaling up. It is nice that the authors provide hyperparameter check for d in Figure 5, and show that d can be selected based on the eigen gap. But the experiment is only done in one pixel-input task Breakout (which is a relatively simple game), making the results less convincing. More experimental results would be appreciated.

The explanation of the sample complexity is too simplified. How would the depencence on $d, B, L_E, K, \Sigma$ influence the complexity? They seem to reveal the intrinsic hardness of defending in the enviornment.

---

> ### Author Response · Authors · 2022-08-02
> **Response to Official Review of Paper12310 by Reviewer uJFB**
>
> We thank the reviewer for their valuable feedback. Please find our response to the raised questions and comments below.
>
> Q1. What is the $K$ in the sample complexity result?
> - $K$ is the sub-gaussian coefficient of the state occupancy distribution as described in the appendix  lemma 6. We apologise for missing it in the theorem 4 and will include this in the final version.
>
> Q2. What is the computational complexity of the algorithm in theory and in experiments?
> - The computation complexity of the algorithm is $O(D^{3})$ and an additional $O(D^2)$ compute time is required for projections at each time step of the environment execution. One need to do SVD only once and later use the recovered subspace for projection at each time step. One time SVD costs $O(D^{3})$, and projection at each time step cost $O(D^2)$ because of matrix vector multiplication.
>
> W1. Difficulty with scaling up. Experiment is only done on one pixel input task?
> - The main step in algorithm requires doing an SVD and doing projection at each time step both of which scale atmost to order 3 in dimension. We have also provided experimental results on two different environments with different types of triggers, specifically the trigger in breakout game is a multi-pixel patch based trigger.
>
> W2. The explanation of the sample complexity is too simplified. How would the depencence on $d, B, L_E, K, \Sigma$ influence the complexity?
> - The sample complexity dependence on these parameters come up naturally from the concentration argument of covariance matrix of sub-gaussian distribution(lemma 6, 7) in the analysis.

---

### Official Review · Reviewer_JEuN · 2022-07-11

**Rating:** 6
**Confidence:** 4
**Soundness:** 3 good
**Presentation:** 3 good
**Contribution:** 3 good

**Summary:**

Trustworthy AI is of interest to many as systems are quickly being deployed in the real world. There is a chance that the models being deployed may contain a backdoor policy which acts maliciously when triggered by an interested attacker who may perturb the agents observations. This work attempts to clearly define the objective of such an attacker, describes the limits of their behavior and attempts to defend against such an attack. It introduces the concept of a safe subspace based on testing the provided policy on a clean environment and then uses this empirically derived subspace to sanitize the policy (akin to a 'denoising' step). Bounds on performance of this sanitized policy w.r.t. the optimal policy are given.

**Questions:**

1. Is it right to say that to get the safe subspace, the agent must sample episodes in an environment free of the trigger? (Pg. 4 :L120)
2. To get the sanitation policy, $n$ clean episodes are sampled and 1 state is sampled from each episode. How is this more helpful than choosing correlated samples (say averaging over $n$ entire episodes) ? (Pg 8 :L211)
3. To get a good value of $d$ when it is unknown to the defender, tests will be need to run on the triggered environment (in the presence of the attacker). The experiments in Fig. 5 (Sec 5.4) seem to be testing using a fixed $n=40$. Why is this so low when Breakout clearly needs a large number of samples to approximate the safe subspace (Fig. 4)?
4. How are the backdoor policies trained and the trigger step decided? Is the trigger action hardcoded into the policy? (Sec 5.1, 5.2)
5. Possible Typos?

    Pg 2
    L46 (which uses autoencoder)
    L51 (have also studies)

    Appendix pg 14 L369 ($s_t$ should be $s$)

**Limitations:**

Point on attacker needing to limit in $E^{\perp}$ space (may not be true in practice) and need for clean environment interactions to get sanitization policy is mentioned. It is not certain that this will generalize to continuous action spaces.


**Strengths And Weaknesses:**

Strengths:
+ Intuitive examples (Fig.2) explaining role of backdoor attacker
+ Bounds provided for sanitized policy behavior in clean environment
+ Tackles question of choice when subspace dimension not known (Sec. 5.4)

Weaknesses:
- Experiments on discrete action-spaces, does this generalize to Mujoco environments and robotic problems?
- Analysis is on zero mean state case and claims to generalize to the non-zero mean case, but this is not substantiated. (in my view)
- Assumptions on attacker might not be valid in many cases (limiting to $E^{\perp}$)

---

> ### Author Response · Authors · 2022-08-02
> **Response to Official Review of Paper12310 by Reviewer JEuN**
>
> We thank the reviewer for their valuable feedback. Please find our response to the raised questions and concerns below.
>
> Q1. Is it right to say that to get the safe subspace, the agent must sample episodes in an environment free of the trigger?
> - Yes, that is correct. Note that with only a few samples in the $E^\perp$ subspace, the adversary can make the trigger direction to be a high variance direction making it very hard for the defender to recover the true (high variance) safe subspace using the eigen values estimated from poisoned samples.
>
> Q2. To get the sanitation policy, clean episodes are sampled and 1 state is sampled from each episode. How is this more helpful than choosing correlated samples (say averaging over entire episodes)?
> - From theory perspective, we need independent samples from the underlying state occupancy distribution $d_\mu^{\pi^*}$ to make sure that the empirical estimates of the projection matrix is unbiased and consistent. This translates to sampling a trajectory and then selecting one of the state in the trajectory uniformly. Practically speaking, it is fine to use correlated samples from a trajectory(this is also what we have done in the experiments) since we are just concerned with recovering the safe subspace defined with respect to the optimal state occupancy distribution. In the worst case though, the states could be very correlated(say all state in a trajectory are same) and we may not get any advantage by using multiple states from a trajectory.
>
> Q3. The experiments in Fig. 5 (Sec 5.4) seem to be testing using a fixed $n=40$ . Why is this so low when Breakout clearly needs a large number of samples to approximate the safe subspace (Fig. 4)?
> - We would like to apologise for missing a line here.  We have actually used $n=40$ for boxing-ram game and $n = 2^{15} = 32768$ for breakout game.
>
> Q4. How are the backdoor policies trained and the trigger step decided? Is the trigger action hardcoded into the policy? (Sec 5.1, 5.2)
> - The backdoor pocily has been trained using a reward poisoning schema as presented in the original TrojDRL [1] paper. The trigger behaviour is not harcoded and is eventually learned by the agent through the poisoned reward signal recieved from the environment during training. We have used targeted attack in breakout game and untargeted attack in the boxing ram game.
>
> Q5. Possible Typos?
> - Thank you for pointing the typos. We will fix them in the paper.
>
>
> W1. Experiments on discrete action-spaces, does this generalize to Mujoco environments and robotic problems?
> - Our sanitizaiton algorithm works by projecting the triggered state onto the safe state subspace and feeding back the projected state as input to the backdoor model to get next action distribution. Since the projection is entirely state dependent, it would directly work with continuous control environments as well.
>
> W2. Analysis is on zero mean state case and claims to generalize to the non-zero mean case, but this is not substantiated. (in my view)
> - We note that estimating mean $\mu$ is much less costly than estimating eigen spectrum of a covariance matrix so as such generalizing the algorithm to non-zero mean would just introduce addition lower order error term due to estimation of $\hat \mu$. We have choosen $\mu=0$ just for ease of exposition and to keep the analysis clean. We will include the details about the non-zero case in the supplement of final paper.
>
> References :
> [1]. TrojDRL: Evaluation of Backdoor Attacks on Deep Reinforcement Learning, Panagiota Kiourti and Kacper Wardega and Susmit Jha and Wenchao Li, DAC 2020.

---

### Meta-Review · Area_Chair_2BFz · 2022-08-24

**Recommendation:** Accept
**Confidence:** Certain

**Metareview:**

The authors present a novel algorithm for defending against backdoor policies in Reinforcement Learning (RL).
The main idea is to project observations onto a "safe" subspace which cleans out the backdoor.

The authors present both empirical finds and theoretical results for their method.
There was an active discussion between reviewers and authors in which the main concerns were addressed.

I recommend that the authors follow the suggestion of reviewer iqtz and emphasise the restriction on the attacker ability more clearly to avoid any over-claims about the capability of the defense method.

**Award:**

No

---

### Decision · Program_Chairs · 2022-09-14

Accept